# MiRNA-Based Therapies for Lung Cancer: Opportunities and Challenges?

**DOI:** 10.3390/biom13060877

**Published:** 2023-05-23

**Authors:** Han Yang, Yufang Liu, Longqing Chen, Juanjuan Zhao, Mengmeng Guo, Xu Zhao, Zhenke Wen, Zhixu He, Chao Chen, Lin Xu

**Affiliations:** 1Special Key Laboratory of Gene Detection and Therapy of Guizhou Province, Zunyi Medical University, Zunyi 563000, China; 2Department of Immunology, Zunyi Medical University, Zunyi 563000, China; 3Institute of Biomedical Research, Soochow University, Soochow 563000, China; 4Collaborative Innovation Center of Tissue Damage Repair and Regeneration Medicine of Zunyi Medical University, Zunyi 563000, China

**Keywords:** microRNAs, lung cancer, chemoradiotherapy, targeted therapy, immunotherapy, MiRNA-based therapy

## Abstract

Lung cancer is a commonly diagnosed cancer and the leading cause of cancer-related deaths, posing a serious health risk. Despite new advances in immune checkpoint and targeted therapies in recent years, the prognosis for lung cancer patients, especially those in advanced stages, remains poor. MicroRNAs (miRNAs) have been shown to modulate tumor development at multiple levels, and as such, miRNA mimics and molecules aimed at regulating miRNAs have shown promise in preclinical development. More importantly, miRNA-based therapies can also complement conventional chemoradiotherapy, immunotherapy, and targeted therapies to reverse drug resistance and increase the sensitivity of lung cancer cells. Furthermore, small interfering RNA (siRNA) and miRNA-based therapies have entered clinical trials and have shown favorable development prospects. Therefore, in this paper, we review recent advances in miRNA-based therapies in lung cancer treatment as well as adjuvant therapy and present the current state of clinical lung cancer treatment. We also discuss the challenges facing miRNA-based therapies in the clinical application of lung cancer treatment to provide new ideas for the development of novel lung cancer therapies.

## 1. Introduction

Lung cancer is the leading cause of cancer-related death worldwide and is also one of the most serious malignant tumors, endangering human health and lives. Histologically, lung cancer can be classified as small-cell lung cancer (SCLC) or non-small-cell lung cancer (NSCLC). The latter accounts for more than 76% of lung cancer cases [1], and this paper mainly discusses NSCLC. Furthermore, there were 1.28 million new NSCLC cases recorded from 2010 to 2017 in the United States [2]. Traditional treatments for NSCLC, including surgery, radiotherapy, and chemotherapy, are often ineffective for patients with advanced NSCLC. Although new progress has been made in immune checkpoints and targeted therapies in recent years, the prognosis of NSCLC patients is still poor; the median overall survival (OS) of advanced NSCLC patients has only increased by 1.5 months in the United States [3]. The treatment of lung cancer still faces numerous challenges due to drug resistance after conventional chemotherapy [4], recurrence and the generation of cell resistance after radiotherapy, the acquisition of resistance-conferring genetic mutations caused by targeted therapy [5], and related immune adverse responses following immunotherapy [6]. Furthermore, micorRNA (miRNA)-based therapies have shown great potential to become novel therapies for NSCLC. Therefore, miRNA-based therapy is likely to be a very promising new treatment strategy (Figure 1). 

MiRNAs are a family of short noncoding RNAs that are widely distributed in eukaryotes and are single-stranded RNAs consisting of 19–25 nucleotides. MiRNA expression is highly conserved and displays high sequentiality (stage specificity) and tissue specificity. In general, miRNAs regulate gene expression through three mechanisms: cleavage of mRNA to promote mRNA degradation, inhibition of mRNA translation, and miRNA-mediated mRNA deadenylation and decay [15]. Studies have shown that the expression of some miRNAs is dysregulated in NSCLC. Moreover, these small molecules influence almost every genetic pathway, from cell cycle checkpoints and cell proliferation to apoptosis, and they regulate a wide range of target genes (Figure 2). It has been estimated that miRNAs can target more than 60% of human protein-coding genes [16]. Currently, studies indicate that more than 2000 miRNAs regulate the expression of approximately one-third of genes in the human genome [17]. Thus, supplements or inhibitors of miRNAs can directly modulate the biological behavior of lung cancer cells and control the progression of cancer. In addition, miRNA-based therapies can be applied as complementary strategies to chemotherapy to reverse drug resistance [4]. Furthermore, they can be combined with radiotherapy to improve the radiosensitivity of lung cancer cells [18] and with immunotherapy to improve the sensitivity, adaptability, and effector function of T cells to tumor antigens [19]. Additionally, miRNA-based therapies can be used to reverse resistance to epidermal growth factor receptor (EGFR)-tyrosine kinase inhibitors (TKIs) as an alternative treatment [20]. Thus, miRNA-based therapies can improve the therapeutic efficacy of lung cancer.

However, to date, only a few miRNA-targeting drugs have entered clinical trials, and most of the relevant studies are still being conducted at the cellular level and in animal testing. Currently, there are still many challenges, such as complex regulatory networks, the efficacy and safety of the drug, drug resistance, and poor delivery systems. It is urgent to further address these challenges of miRNA-based treatment for NSCLC. This article reviews the recent progress and potential value of miRNAs for NSCLC treatment and adjuvant existing NSCLC therapies based on the current status of NSCLC clinical treatment, with the aim of providing new ideas for the further development of miRNA-based treatment strategies for NSCLC.

## 2. MiRNA-Based Lung Cancer Therapies

Cancer-related miRNAs can be broadly divided into two categories: tumor suppressor gene miRNAs and oncogene miRNAs [21]. The function of miRNA genes depends on their target in a particular tissue. If the key target of the miRNA gene in a specific cell type is an oncogene, the miRNA gene can be considered a tumor suppressor gene; if the target of the miRNA gene is a tumor suppressor gene in different cell types, the miRNA gene can be considered an oncogene [22]. When multiple miRNAs are overexpressed or inhibited, they can affect the growth of cancer cells in vivo or in vitro. Therefore, we can control the growth of cancer cells through the regulation of miRNAs.

Some miRNAs that act as tumor suppressors are usually downregulated or absent in cells. MiRNA replacement therapy is based on delivering exogenous miRNAs to patients to regulate abnormal functions [21]. For instance, let-7 is highly expressed in normal lung tissue and downregulated in lung cancer cells [23]. Let-7a inhibits NIRF expression in lung cancer while increasing p21^WAF1^ expression and inhibiting cell proliferation [24]. In addition, let-7a inhibits the growth of lung cancer by inhibiting the expression of k-Ras and c-Myc [25]. Similarly, let-7b-3p inhibits the proliferation and metastasis of LUAD cells in vivo and in vitro by directly targeting the BRF2-mediated MAPK/ERK pathway [26]. Let-7c inhibits the migration and invasion of lung cancer by acting on ITGB3 and MAP4K3 targets [27]. Although the effect of a single miRNA acting on a single target may be subtle, the collective suppression of tens to hundreds of genes may have a significant impact on cells and produce strong phenotypic results [28]. Therefore, regulation of the let-7 miRNA tumor suppressor gene shows potential as an alternative therapy tool. Phong Trang et al. found that in a xenotransplantation model of lung cancer, intratumoral injection of let-7 could reduce tumor size, interfere with the carcinogenic characteristics of tumor cells, and induce a therapeutic response [29]. At the same time, Aurora et al. found that nasal injection of let-7 could reduce the formation of lung tumors in animals expressing the G12D activation mutation of the k-Ras oncogene [10]. Therefore, Let-7-based therapies are promising treatment strategies for lung cancer patients. Regulating Let-7 expression may affect tumor proliferation by targeting various oncogenes and inhibiting key regulators of several mitotic pathways [30]. Although the expression of miRNA target genes may vary in different tissues and cells, the ability of miRNAs to target multiple key oncogenes may be more effective than drugs targeting a single gene, which makes miRNAs an attractive therapeutic tool.

Some miRNAs expressed in tumor cells are upregulated. MiRNA inhibitors are natural or artificial RNA transcripts that can isolate miRNAs and reduce or even abolish their activity. Suppressive miRNA therapy is based on alleviating abnormalities in patients by treating them with exogenous miRNA mimics for miRNA inhibition, which include natural miRNA inhibitors such as competitive endogenous RNAs (ceRNAs) and artificial miRNA inhibitors such as anti-miRNA oligonucleotides and miRNA sponges [31,32]. Competitive endogenous RNAs are RNA transcripts, such as mRNAs, lncRNAs, circRNAs, and pseudogene transcripts, that can regulate each other by competing for the same but limited pool of miRNAs to reduce miRNA availability and mitigate repression of target genes [33]. LINC00336, a nuclear lncRNA, is upregulated in lung cancer and functions as an oncogene by acting as a ceRNA. LINC00336 binds to the RNA-binding protein ELAVL1 through nucleotides 1901–2107 of LINC00336 and the RRM interaction domain and key amino acids of ELAVL1, inhibiting ferroptosis. Moreover, ELAVL1 was shown to increase LINC00336 expression by stabilizing it at the posttranscriptional level. LINC00336, as a ceRNA of miR-6852, regulates the expression of the iron death surrogate marker cystathionide-β-synthase (CBS), eventually promoting ferroptosis to inhibit the growth of NSCLC cells [34]. Anti-miRNA oligonucleotides (ASOs) have been developed and used to inhibit miRNAs by directly binding small RNA molecules in the RNA-induced silencing complex (RISC), blocking or inhibiting the ability of endogenous target genes, leading to the translation escalation of miRNA regulatory genes [35]. To increase the binding affinity, specificity, biological stability, and pharmacokinetic properties of targeted miRNAs, effective silencing of dysregulated miRNAs requires chemical modification with anti-miRNA oligonucleotides. The most commonly used sugar modifications to improve the melting point of double-stranded DNA and the resistance to miRNA nuclease include 2’-O-methyl, 2’-O-methoxyethyl, 2’-fluoro, locked nucleic acid modification, and phosphorus thiophosphate modification [36]. Liang et al. found that most miR-21-5p isolated from NSCLC tissues had a 3’-terminal 2’ OME, and this phenomenon was not observed for miRNAs in noncancerous lung tissues. Methylated miR-21-5p was more resistant to digestion by 3’→ 5’ exo-ribonuclease polyribonucleotide nucleotransferase 1 and had a higher correlation with Argonaute-2, which may have contributed to its higher stability and stronger inhibition of programmed cell death protein 4 (PDCD4) [37] (Table 1). MiRNA-based therapies for NSCLC, which are an extremely promising new therapy, can directly affect a variety of biological behaviors of lung cancer cells to control the development of lung cancer.

## 3. MiRNAs as Complementary Therapies

Traditional treatments for NSCLC include surgery, radiotherapy, and chemotherapy. Patients with early stages of NSCLC (stage I, stage II, and stage IIIA) are mainly treated by radical surgical resection, with adjuvant cisplatin-based chemotherapy for those patients in stages II-IIIA. Unfortunately, the OS for patients with advanced NSCLC receiving combination chemotherapy is less than 2 years [38], the risk of death is only reduced by 5.4% at five years, and there is a high recurrence rate and a relatively high level of drug toxicity [39]. Compared to a decade ago, significant advancements in the clinical treatment of lung cancer have been realized due to the application of targeted therapy and immunotherapy [40]. Most stage III NSCLC patients are nonsurgical candidates, and the current standard of care is concurrent chemoradiotherapy followed by immunotherapy [41]. Targeted therapy for patients with alterations in EGFR, ALK, ROS1, RET, BRAF V600E, MET exon 14, and NTRK genes can greatly improve clinical outcomes [42]. Although the median survival of NSCLC patients has been prolonged after the application of targeted therapy and immunotherapy in clinical therapies, there are still many challenges in the treatment of NSCLC [43,44]. Recent evidence has shown that miRNA-based therapy might be a valuable complementary strategy to increase the cure rate of lung cancer patients.

### 3.1. MiRNAs and Chemotherapy

Cisplatin (CP), which combines with DNA to induce cross-linking, destroys the function of DNA, and inhibits cell mitosis, is a cell-nonspecific drug that is often used as a chemotherapy agent [4]. However, most patients develop resistance to chemotherapy, and resistance to conventional therapies is one of the major reasons for cancer chemotherapy failure. Cancer cells acquire resistance to chemotherapy through four main mechanisms: (1) abnormal repair of DNA damage in tumor cells can induce the activity of the DNA repair system, leading to the development of chemotherapy resistance in tumor cells [45]; (2) removal of biomolecules or organelles damaged during chemotherapy by autophagy can allow tumor cells to avoid further cell death [46]; (3) abnormal expression of drug transport proteins can lead to the reduction of drug concentration in tumor cells [47]; (4) abnormal expression or defective function of apoptosis-related proteins can lead to apoptosis resistance in tumor cells, which in turn leads to chemoresistance [48]. 

Fortunately, it has been demonstrated that, through their complex underlying regulatory mechanisms, many miRNAs play important roles in the development of chemoresistance (Figure 3). MiR-149 resensitizes drug-resistant NSCLC cells to CP by targeting ERCC1, which is a key gene that promotes abnormal DNA damage repair [49]. MiR-488 induces CP resistance in NSCLC cells by promoting XPC expression through regulation of the elF3a/NER pathway [50]. MiRNAs can also mediate cellular drug resistance by regulating key genes or signaling pathways in autophagy. MiR-23a effectively reverses drug resistance in NSCLC by targeting AKT to inhibit autophagy and induce apoptosis [51]. In addition, miR-138-5P inhibits the extension of autophagosomes to reverse chemotherapy resistance [52]. MiR-124 and miR-142 enhance the sensitivity of NSCLC cells to CP by targeting SIRT1 to inhibit autophagy [53]. Conversely, miR-1269b enhances the cleavage of the autophagosome through the regulation of the PTEN/PI3K/AKT signaling pathway, leading to CP resistance [54]. Furthermore, miRNAs can modulate the expression of various transporter proteins, thereby reversing CP resistance in NSCLC cells. For instance, hsa-miR-98-5p inhibits the expression of copper transporter 1 (CTR1) to reverse CP resistance in NSCLC [55]. By targeting the miR-361-3p/ABCC1 axis, circ0058357 inhibition reduces the proliferation and metastasis of NSCLC cells and partially sensitizes them to CP [56]. In addition, miR-1914-3p can also reverse chemoresistance by regulating yes-associated protein (YAP) [57]. A variety of abnormally expressed miRNAs in NSCLC cells have been shown to be associated with escape from cell death. MiR-137 increases reactive oxygen species (ROS) production, reduces GSH and SOD levels, and leads to apoptosis in NSCLC cells [58]. Exosome-transmitted circVMP1 promotes NSCLC progression and CP resistance by targeting the miR-524-5p-METTL3/SOX2 axis [59]. MiR-125b-5p increases CP resistance in NSCLC in vivo and in vitro by targeting CREB1 to inhibit apoptosis [60]. In addition, miR-744-5p/miR-615-3p inhibit GPX4-mediated iron death and reverse CP resistance in NSCLC [61]. Furthermore, miRNAs such as miR-4333, miR-522, and miR-362-3 can also regulate ferroptosis to influence chemoresistance [62,63,64] (Figure 3).

Furthermore, oncogenic miRNAs trigger CP resistance in NSCLC cells by targeting various pathways, such as Wnt/β-catenin, Rab6, CASP2, PTEN, and Apaf-1. MiR-181a induces lung cancer cells to acquire CP resistance by increasing PD-L1 expression [65]. MiR-1269b targets PTEN to modulate the PI3K/AKT signaling pathway and drives CP resistance in human NSCLC cells [54]. MiR-324-5p enhances resistance to CP by targeting FBXO11 in NSCLC cells [66]. In contrast, tumor suppressor miRNAs inhibit oncogene pathways such as STAT3, LRP8, and TACC1 to suppress CP resistance. MiR-200c-3p and miR-485-5p were both downregulated and targeted RRM2 in CP-resistant NSCLC tissues and cells. Overexpressing miR-200c-3p or miR-485-5p suppressed the CP resistance and malignant behaviors of NSCLC cells [67]. MiR-526b-3p activated CD8^+^ T cells in a STAT3/PD-L1-dependent manner, reversed CP resistance, and suppressed metastasis in NSCLC cells [68]. MiR-30b-5p resensitized NSCLC cells to CP by targeting LRP8, and cell viability, migration, invasiveness, and tumorigenesis were significantly diminished after miR-30b-5p overexpression, while cell apoptosis rates were increased [69]. A recent study showed that miR-193a in exosomes suppressed colony formation, invasion, migration, and proliferation as well as increased apoptosis of NSCLC CP-resistant cells by targeting LRRC1 [70]. These studies suggest that miRNA regulation may overcome the increasingly widespread mechanisms of drug resistance in the field of oncology and propose its use in therapeutic approaches for targeting tumors with multi-drug resistance phenotypes.

### 3.2. MiRNAs and Radiotherapy

Radiotherapy (RT) is an adjunct to surgery and is a therapeutic choice mainly for patients with regionally unresectable advanced lung cancer [71]. Although extensive technical advancements in RT have been made in recent years, a significant proportion of patients still present with tumor recurrence. Although the RT dose is well standardized among patients, isolated local recurrences can occur; even in the modern era of dose escalation, posttreatment biopsies show a 15–20% residual disease rate [72]. Ionizing radiation damages cells by producing intermediate ions and free radicals that cause DNA single-strand breaks (SSBs) or double-strand breaks (DSBs), while DSBs are the more lethal type of injury. Radiation-induced DSBs trigger a DNA damage response, comprising a network of proteins affecting DNA repair and signaling [73]. Ionizing radiation-induced DNA damage triggers signals that can eventually activate temporary checkpoints for gene repair or irreversible growth arrest, leading to necrosis or apoptosis. Such checkpoint activation constitutes an integrated response that involves sensor (RAD, BRCA, NBS1), transducer (ATM, CHK), and effector (p53, p21, CDK) genes [74]. It has been shown that overexpression of miR-34 inhibits the efficiency of homologous recombination repair and induces DSBs through the downregulation of RAD51 expression [75]. Similarly, miR-124 [76], miR-205-5p [77], and miR-193a-3p [78] can also influence the efficacy of radiotherapy by affecting DNA repair. 

Furthermore, MiRNAs can cause abnormalities in the cell cycle by regulating different phases of the cell cycle, which in turn promotes or diminishes the efficacy of radiotherapy. For example, miR-519a arrests the cell cycle in G0/G1 by inhibiting the expression of the EphrinA2 receptor (EphA2) [79]. MiR-195-5p targets HOXA10, leading to G1 phase block and apoptosis [80]. MiR-30a induces G2/M checkpoint block by targeting activating transcription factor 1 (ATF1) [81]. In addition, miR-365 affected the entire cell cycle by down-regulating CDC25A and increasing the sensitivity of NSCLCs to radiotherapy [82]. Hypoxia plays an important role in tumor resistance to radiotherapy by upregulating hypoxia-inducible factors (HIFs) to stimulate enzymes responsible for cancer survival under hypoxic stress [83]. MiR-18a-5p increases radiosensitivity in lung cancer cells and CD133^+^ stem-like cells by downregulating ATM and HIF-1α expression [84]. In addition, both miR-210 [85] and miR-199a-5p [86] can promote chemoresistance by affecting the hypoxic environment induced by HIF-1 (Figure 4).

Xiao et al. found 12 differentially expressed miRNAs in radiotherapy-sensitive and radiotherapy-resistant NSCLC patients, and these miRNAs may be used as candidate markers for radiotherapy sensitivity. Compared with radiotherapy-resistant patients, five miRNAs (miR-126, miR-let-7a, miR-495, miR-451, and miR-128b) were significantly upregulated, and seven miRNAs (miR-130a, miR-106b, miR-19b, miR-22, miR-15b, miR-17-5p, and miR-21) were greatly downregulated in the radiotherapy-sensitive group. Overexpression of miR-126 increased the radiosensitivity of lung cancer cells through regulation of the PI3K-AKT pathway [87]. MiR-296 expression levels decreased, while lncRNA AGAP2 antisense RNA (AGAP2-AS1) expression levels increased in lung cancer cells and tissues. Moreover, the M2 macrophage-derived exosome AGAP2-AS1 enhanced lung cancer radiotherapy resistance by reducing miR-296 and elevating NOTCH2 [88]. MiR-219a-5p enhanced the radiosensitivity of NSCLC cells by targeting CD164 [89]. Recent studies showed that miR-20b-5p increased the sensitivity of lung cancer cells by targeting PD-1. In transplanted tumor-bearing nude mice, either pembrolizumab, a humanized monoclonal anti-PD1 antibody, or miR-20b-5p overexpression enhanced the radiosensitivity of NSCLC cells [90]. In addition, a clinical study showed that miRNAs may serve as valuable biomarkers to predict patients who can benefit from high-dose RT [91]. With advances in the understanding of miRNAs, miRNA-based therapy in conjunction with RT may be a promising strategy to tailor personalized treatment. The ability to tailor miRNA-based applications in the clinical arena brings great hope for radiation oncologists, and this approach can lead us toward personalized lung cancer RT.

### 3.3. MiRNA-Assisted Targeted Therapy

It is well known that conventional radiotherapy causes substantial physical, chemical, and biological damage [92]. Compared to conventional radiotherapy, targeted therapy targets tumor cells at the site of tumorigenesis with a precise and targeted attack, providing greater specificity and efficiency in tumor treatment [93]. In NSCLC, common mutation loci include EGFR [94], allogeneic lymphoma kinase (ALK) [95], mesenchymal-epithelial transition factor (MET) [96], and Kirsten rat sarcoma (KRAS) [97]. The most frequent mutation is in EGFR, and inhibitors targeting the structural domain of the EGFR kinase, EGFR-TKIs, have been developed as the standard first-line treatment for patients with EGFR mutations in advanced NSCLC, achieving excellent results in terms of progression-free survival and OS [98].

However, most patients inevitably develop acquired resistance to targeted therapeutics, and the mechanisms of acquired resistance can be divided into four types: (1) Alterations in EGFR signaling pathways—EGFR tertiary mutations [99] and amplification leading to EGFR-TKI target failure [100]. Exosomal secretion of miR-7 alters the resistance phenotype of gefitinib-resistant cells in vitro by targeting YAP and reversing drug resistance [101]. MiR-34a reverses HGF-induced EGFR-TKI resistance in EGFR-mutated NSCLC cells [102]. In addition, miR-34c, miR-183, and miR-210 are all altered in EGFR mutations; however, the exact mechanism remains to be investigated [103]. (2) Activation of aberrant bypass pathways—human epidermal growth factor receptor 2 (HER2) amplification [104] and aberrant insulin-like growth factor 1 receptor (IGF-1R) activation, which competes with EFGR-TKI for mutual exclusion and attenuates the sensitivity of EGFR-TKI therapy [105]. MiR-4728 affects the efficacy of targeted therapy for HER2-amplified breast cancer by targeting ESR1 and modulating ERα-mediated NOXA transcription to slow apoptosis following treatment with the HER2 inhibitor lapatinib [106]. FOXO3a-miRNA negative feedback inhibition loop affects targeted therapeutic resistance by inhibiting the IGF2/IGF-1R/IRS1 signaling pathway [107]. MiR-30a-5p blocks the PI3K/AKT signaling pathway through dual inhibition of EGFR and IGF-1R to increase targeted therapeutic sensitivity [108]. (3) Activation of the downstream pathways RAS/RAF/MEK/ERK [109] and PI3K/AKT/mTOR [110] reduces cellular sensitivity to EGFR-TKIs. Histone deacetylase (HDAC) inhibitor ITF2357 increases miR-130a-3p expression by inhibiting HDAC2. Thereby, miR-130a-3p targets RAD51 to reduce resistance to pemetrexed in KRAS-mutated NSCLC cells [111]. In addition, miR-202 [112], miR-200c [113], and miR-199a-3p/5p [114] can affect RAS/RAF/MEK/ERK or PI3K/AKT/mTOR signaling pathways through the corresponding target genes, affecting the efficacy of targeted therapies. Additionally, (4) Histological/phenotypic transformation—SCLC transformation [115] and epithelial mesenchymal transition (EMT) [116]. Furthermore, overexpression of miR-625-3p reduced IC50 values in gefitinib-resistant cell lines (HCC827GR). HCC827GR cells transfected with miR-625-3p showed increased expression of E-cadherin and decreased expression of N-cadherin and vimentin. Mechanistically, miR-625-3p overexpression reversed TGF-β1-induced EMT and enhanced gefitinib sensitivity by directly targeting receptor tyrosine kinase (AXL) in lung cancer cells [117]. In addition, ectopic overexpression of miR-506-3p in erlotinib-resistant cells targeted Sonic Hedgehog (SHH), increased E-cadherin expression, inhibited vimentin expression, and reversed EMT to a mesenchymal phenotype epithelial-like transformation, thereby counteracting EMT-mediated chemoresistance, increasing sensitivity to apoptosis, decreasing cell stemness, reducing proliferation, and enhancing sensitivity to erlotinib [118] (Figure 5).

Numerous studies have shown that multiple miRNAs are extensively involved in EGFR-TKI regulation, suggesting that miRNAs may be new therapeutic molecules and biomarkers for anti-EGFR therapy and have a positive role in reversing drug resistance and improving sensitivity [119,120,121]. Alessandro et al. collected plasma samples from 39 patients with advanced EGFR-mutated NSCLC treated with EGFR-TKIs, assessed the expression levels of miRNAs and found that miR-21 could be a useful indicator to monitor the effect of EGFR-TKI treatment [122]. In addition, Zhang et al. identified miR-608 and miR-4513 single nucleotide polymorphisms as independent candidate biomarkers for predicting survival in lung adenocarcinoma patients after EGFR-TKI treatment by systematically screening the database of 1000 genomic projects in miRbase and obtaining data from 319 stage IIIB/IV patients treated with EGFR-TKI [123]. However, the development of resistance, whether emerging or acquired, can limit the efficacy of EGFR-TKIs for long-term use (less than one year). The expression levels of miRNAs upon EGFR-TKI treatment have been reported to differ between drug-resistant and sensitive tumor cells and undergo differential regulation; therefore, miRNA-based therapies may be a possible strategy to reverse drug resistance and present a targeted therapy option for the adjuvant treatment of cancer. For instance, the expression of miR-125a-5p, which exerts its antitumor effects by targeting EGFR, is decreased in human tumor cells [124,125]. Jamal et al. found that miR-125a-5p pretreatment synergistically increased the cytotoxic effects of erlotinib (first-generation EGFR-TKI inhibitors) and decreased the IC50 values of tumor cells, and miR-125a-5p significantly enhanced the apoptotic effects induced by erlotinib [13]. In conclusion, miRNA-assisted targeted therapy to reverse drug resistance and enhance the sensitivity of lung cancer cells to targeted therapies could be a promising treatment option.

### 3.4. MiRNA-Assisted Immunotherapy

The purpose of tumor immunotherapy is to activate the body’s immune system, and it relies on autoimmunity to kill cancer cells and tumor tissues. Unlike surgical, chemotherapy, radiotherapy, and targeted therapy strategies, immunotherapy does not target tumor cells and tissues but instead targets the body’s immune system [126]. A well-studied immune checkpoint is cytotoxic T lymphocyte-associated antigen-4 (CTLA-4), which primarily regulates T-cell activity in the early tumor microenvironment and has a strong correlation with T cells in all cancer types [127,128]. In addition, PD-1 and its ligands, PD-L1 and PD-L2, can limit T-cell activity in the late stages of tumor development and thus exert an immunomodulatory function [129]. Inhibitors targeting the CTLA-4 and PD-1/PD-L1 pathways have been developed to exert antitumor effects. However, immune checkpoint inhibitors (ICIs) provide therapeutic benefits with some degree of resistance and side effects, including cytokine storms, small bowel colitis, and hematologic immune-related adverse events [130,131,132,133]. Therefore, the development of potential modulators of immune checkpoint pathways is important to further enhance the effectiveness of antitumor immunotherapy.

Furthermore, the altered expression of key molecules in a number of signaling pathways plays a key role in immunotherapy resistance in tumor cells. Overactivation of the MAPK signaling pathway leads to the production of IL-8 and VEGF by tumor cells, promoting tumor angiogenesis, growth, and metastasis and inhibiting T lymphocyte infiltration into tumor tissue and resistance to immune checkpoint therapy [134,135]. In NSCLC cells, the lncRNA FGD5-AS1 acts as a sponge for miR-454-3p to upregulate ZEB1 expression, thereby increasing the expression of PD-L1 and VEGFA and promoting NSCLC angiogenesis and immune evasion [136]. The EGFR-P38 MAPK axis can upregulate PD-L1 via miR-675-5p, which enhances PD-L1 mRNA stability and leads to PD-L1 accumulation [137]. Loss of PTEN protein activity in NSCLC leads to increased PI3K activity, which reduces the activity of cytotoxic T lymphocytes and limits their invasion into tumor tissue, in addition to affecting the antitumor effect of INF-γ [138]. MiR-103a decreases PTEN levels in tumor microenvironment monocytes by increasing the polarization of M2 macrophages, which in turn promotes the activation of AKT and STAT3, leading to the development of tumor immunosuppression [139]. In addition, both miR-425 and miR-576 can decrease the activity of PTEN and exert immunosuppressive effects [140]. Overactivation of the Wnt signaling pathway inhibits the production of MIP-1, thereby reducing the infiltration of CD103^+^ dendritic cells (DCs) into the tumor tissue and thus affecting the tumor-killing effect of cytotoxic T lymphocytes [141]. Moreover, it has been shown that miR-18a leads to reduced antigen presentation and thus immune escape in ER-positive breast cancer cells through modulation of Wnt signaling [142]. Mutations of the JAK1/JAK2 gene, found in tumor cells, promote excessive activation of the STAT family, leading to abnormal cell surface IFN-γ receptors, which are insensitive to IFN-γ secreted by cytotoxic T cells [141]. Overexpression of miR-4458 inhibits tumor growth, decreases the proportion of PD-1^+^ T cells and the expression of PD-L1 and IL-10, and upregulates the proportion of CD4^+^ T and CD8^+^ T cells as well as the expression of IFN-γ and IL-2 [14]. In addition, both miR-488 [143] and miR-181a-5p [144] can affect the JAK/STAT signaling pathway, but their effects on immunotherapy need to be further explored (Figure 6).

Studies have shown that the function of miRNAs in lung cancer is closely related to the expression of CTLA-4 and PD-1/PD-L1, while miRNAs play a regulatory role related to immune checkpoints, thus participating in immune responses and immunotherapy [145,146,147]. One study showed that the expression of CD3/CTLA-4 was significantly higher in NSCLC patients, which was positively correlated with miR-146a in patients with the CC genotype; thus, miR-146a could predict the treatment effect of ICIs in patients with advanced NSCLC [148]. MiR-33a was inversely correlated with PD-L-1 and PD-L-1 expression, and patients with high miR-33a/low PD-L-1 expression had a better prognosis, suggesting that miR-33a could be used as a prognostic marker for PD-1 treatment response [149]. Let-7b post-transcriptionally suppressed PD-L1 and PD-1 expression in the tumor microenvironment, and let-7b treatment reduced PD-1 expression in CD8^+^ T cells, decreased PD-L1 expression in lung cancer cells, and enhanced the function of antitumor CD8^+^ T cells [150]. In addition, studies have shown that traditional Chinese medicine and circular RNAs can regulate the expression of miRNAs and the tumor immune response. The expression of circHMGB2 was significantly upregulated in NSCLC tissues, and circHMGB2 relieved the inhibition of downstream CARM1 by sponging miR-181a-5p, thus inactivating the type 1 interferon response in NSCLC tissues. Upregulation of circHMGB2 expression decreased STAT1 phosphorylation and inhibited IFN-γ signaling, driving immunosuppression and anti-PD-1 resistance in NSCLC. A previous study revealed that the combination of CARM1 inhibitors and anti-PD-1 antibodies was a very promising treatment option for NSCLC [144]. Moreover, nobiletin is a natural flavonoid isolated from the citrus peel that has anti-inflammatory and anticancer functions. Nobiletin could inhibit PD-L1 expression by regulating miR-197, thereby avoiding the occurrence of immune escape in NSCLC [151]. 

In all, the introduction of miRNA therapies, which have been combined with immunotherapy to modulate immune checkpoint expression and enhance T-cell sensitivity, can further enhance the antitumor effect.

## 4. Clinical Trials Based on miRNAs

Let-7, the first human miRNA to be discovered, was shown to inhibit tumor growth in a mouse model back in 2008 [10]. Over the following decade, an increasing number of miRNAs have demonstrated their anti-NSCLC effects in animal studies, and delivery systems have become more diverse. Delivery of miR-29b and miR-200c via liposomes significantly inhibited the proliferation and increased radiosensitivity of NSCLCs [152,153]. Subsequent studies have also shown that nanodelivery of miR-34a and let-7b can inhibit the growth of NSCLCs, and this combined delivery of miRNAs has also shown a better survival advantage [28]. A recent study found that a 231-exosome loaded with miR-126 specifically identified NSCLCs in peripheral blood and blocked their proliferation and metastasis [154]. This organotropism feature of exosomes may provide better safety and efficacy when administered systemically. Meanwhile, loading miR-101 by extracellular vesicle transfer also showed promising anti-tumor effects [155]. Ultrasound-targeted microbubble destruction (UTMD) delivery of miR-21-5p inhibitors reduced the size of xenograft tumors, and UTMD transfection was more effective than that of liposomes [156]. Although miRNAs have shown potent anti-NSCLC effects, most of the relevant studies are still being conducted at the cellular level and in animal testing. 

A clinical trial on the use of TargomiRs, which are composed of miR-16 mimics, nanoparticles, and a targeting component, for the treatment of lung cancer has been completed (NCT02369198). The miR-16 family has been implicated as a tumor suppressor in a range of cancer types. MiR-16 expression is reduced in lung cancer and regulates the ERK/MAPK signaling pathway to inhibit proliferation and invasion by targeting MAPK kinase 1 (MEK1) [157]. Additionally, miR-16 is a master regulator of the fibroblast secretome, and its upregulation reduces HGF secretion by fibroblasts, impairing their capacity to promote cancer cell migration [158]. MiR-16 directly targets the three KRAS downstream effectors MAPK3, MAP2K1, and CRAF in NSCLC, restoring sensitivity to erlotinib in KRAS-mutated NSCLC both in vitro and in vivo. This previous study also provided evidence that the miR-16—erlotinib regimen is more effective than the selumetinib—erlotinib combination in KRAS-mutated NSCLC [159]. The study enrolled 27 participants over 18 years of age with malignant pleural mesothelioma and non-small cell lung cancer. Patients were given TargomiRs via 20 min intravenous infusion either once or twice a week (3 days apart) in a traditional 3 + 3 dose-escalation design in five dose cohorts. Of the 22 patients who were assessed for response by CT, one (5%) had a partial response, 15 (68%) had stable disease, and six (27%) had progressive disease. The proportion of patients who achieved an objective response was therefore one (5%) of 22, and the duration of the objective response in that patient was 32 weeks. The median overall survival was 200 days (95% CI 94–358). During the trial, 21 deaths occurred, of which 20 were related to tumor progression and one was due to bowel perforation [160].

A recent phase 1 study using liposomal miR-34a mimics in patients with advanced solid tumors resulted in the deaths of four patients due to serious immune adverse events and therefore ended the trial (NCT01829971). MiR-34 has been reported to be dysregulated in various human cancers and is regarded as a tumor-suppressive miRNA because of its synergistic effect with the well-known tumor suppressor p53 [161]. MiR-34a and miR-34c, by targeting PDGFR-α and PDGFR-β, increase TRAIL-induced apoptosis and decrease the invasiveness of lung cancer cells [162]. Hong et al. found that miR-34b-3p inhibits lung cancer cell proliferation and is mediated by cell cycle arrest and apoptosis with CDK4 interference [163]. Andrea et al. examined the therapeutically resistant Kras^LSL-G12D/+^ and Trp53^LSL-R172H/+^ mouse lung cancer models and characterized tumor progression in these mice following lung-specific transgene activation. They found tumors as early as 10 weeks postactivation and severe lung inflammation by 22 weeks. In the presence of exogenous miR-34, epithelial cells derived from these tumors show reduced proliferation and invasion. In vivo treatment with miR-34a prevented tumor formation and progression in Kras^LSL-G12D/+^ and Trp53^LSL-R172H/+^ mice [164]. Furthermore, in situ hybridization staining of pre- and posttreatment liver biopsies from patients with various tumor types showed increased miR-34a in tumor tissues following MRX34 treatment, localized to the cellular cytoplasm, verifying delivery of miR-34a to the tumor microenvironment. However, this first-in-human clinical trial of a miRNA-based therapy was halted in 2016 due to unexpectedly severe immune-mediated toxicities, which resulted in four patient deaths in expansion cohorts [12]. In conclusion, miRNAs have shown excellent modulation of NSCLC pathogenesis in these clinical trials. However, the efficient and safe delivery of nucleic acids remains the Achilles heel of gene therapy. Therefore, we need to continue to explore the direction and method of using miRNAs to treat lung cancer (Table 2).

## 5. Conclusions and Perspectives

Increased research on miRNAs, which have shown great potential in the diagnosis and treatment of NSCLC, has contributed to the development of new therapies for NSCLC. Due to the powerful regulatory functions of miRNAs, mimics, and inhibitors of miRNAs play an important role in the treatment of NSCLC as well as other diseases. In addition, miRNAs can also modulate key signaling pathways in NSCLC to reverse drug resistance and improve drug sensitivity, thereby enhancing the efficacy of existing therapies, including adjuvant conventional therapies as well as immunotherapy and targeted therapies. Three studies using small interfering RNA (siRNA) and miRNA mimetic therapeutics to target cancer-related genes have recently received approval from the Food and Drug Administration [165].

Even though a growing number of studies have confirmed the great potential of miRNAs in the treatment of NSCLC, it must be stated that there are still many challenges to overcome before translating these studies into clinical applications (Figure 7). For instance, the optimal selection of miRNAs is critical for the application of miRNA-based therapeutic strategies in lung cancer. In the tumor microenvironment, there is heterogeneity in the expression of miRNAs, leading to difficulties in the identification of target miRNAs [166]. Hypoxia and inflammation, for example, both lead to complex and regionally heterogeneous expression of miRNAs [167,168]. In our previous studies, we used the promoter of TTF-1, the distinct biomarker for NSCLC, to enforce miR-7 expression to reduce the progression of NSCLC [169]. Moreover, optimization of the TTF-1 promoter more effectively inhibited the growth of NSCLC through manipulation of miR-7 [170], indicating the value of the specific promoter-operating expression in miRNA-based therapeutics. Even though advanced technologies such as single-cell multiomics sequencing and high-resolution spatialomics are still needed to be conducted in the future to further explore the heterogeneity of miRNA expression in tumors, it will also be helpful to take multiple biopsies in the progression of lung cancer.

Furthermore, immunotoxic reactions and poor delivery systems are still issues that need to be urgently addressed. Therefore, the development of new delivery strategies is necessary for the drug development of miRNAs. Traditional local delivery systems can inhibit tumor tissue with low toxicity, but because this strategy uses direct injection or local delivery, it is not considered ideal for the treatment of advanced metastatic cancers [171]. Liposomes are one of the most commonly used methods of nonviral vector delivery systems, and a recent phase 1 study using liposomal miR-34a mimics in patients with advanced solid tumors resulted in the deaths of four patients due to serious immune adverse events and therefore ended the trial [12]. This result demonstrates the deficiencies of conventional delivery systems leading to systemic immune activation and that effective delivery remains an unresolved challenge. Recent studies have found that exosomes have excellent biological properties, including biocompatibility, stability, low toxicity, and proficient exchange of molecular cargo, making them prime candidates for drug delivery [172]. Newly developed exosome-/liposome-based nanovesicles are engineered exosome mimics in which the desired components of a natural exosome are incorporated into synthetic liposomes or nanoparticles and assembled using a controlled procedure [173]. Of note, nanovesicles can encapsulate large plasmids or mRNA transcripts while retaining the benefits of exosomes, such as biocompatibility and targeting ability, and can significantly improve disease treatment. This could be a promising drug delivery approach for the treatment of cancer, reducing toxicity and inhibiting metastasis mainly in the lungs [174].

Meanwhile, recent studies have found that the mechanisms of abnormal expression and regulation of miRNAs in tumors also deserve attention. Abnormal expression of miRNAs in tumor tissues in NSCLC might be associated with genomic mutations, DNA methylation, RNA modification, etc. [175]. However, given the complexity of the tumor microenvironment (TME), the exact mechanisms of different miRNAs in distinct types of cells in the TME are still largely unknown. For instance, our previous studies have revealed that the abnormality of miR-7 in NSCLC is related to site mutations in the promoter of miR-7 [176]. However, in the T cell population, miR-7 expression is manipulated by the transcriptional factor C/EBP [177], indicating the diversity of distinct miRNA expression regulation in different cell populations. Therefore, further studies on the mechanisms of abnormal expression of miRNAs in TME are valuable for the elucidation of the etiology of lung cancer and the subsequent development of miRNA-based therapies.

In summary, advances in miRNA-based therapies hold great promise for individualized NSCLC treatment in the future. Findings from these studies, including the mechanisms of abnormal expression, the optimal selection of miRNAs, novel delivery systems, etc., will facilitate the development of low-toxicity, highly effective, and highly targeted miRNA therapies that exploit the powerful regulatory functions of miRNAs to suppress tumor development and ultimately benefit the clinical outcome of NSCLC patients.

## Figures and Tables

**Figure 1 biomolecules-13-00877-f001:**
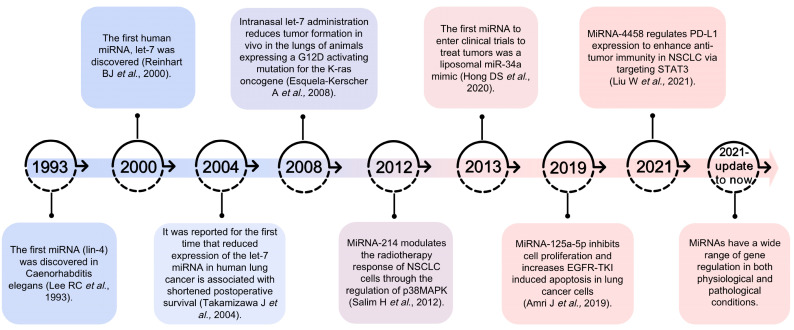
Overview of the progression of miRNA-based therapy research [7,8,9,10,11,12,13,14].

**Figure 2 biomolecules-13-00877-f002:**
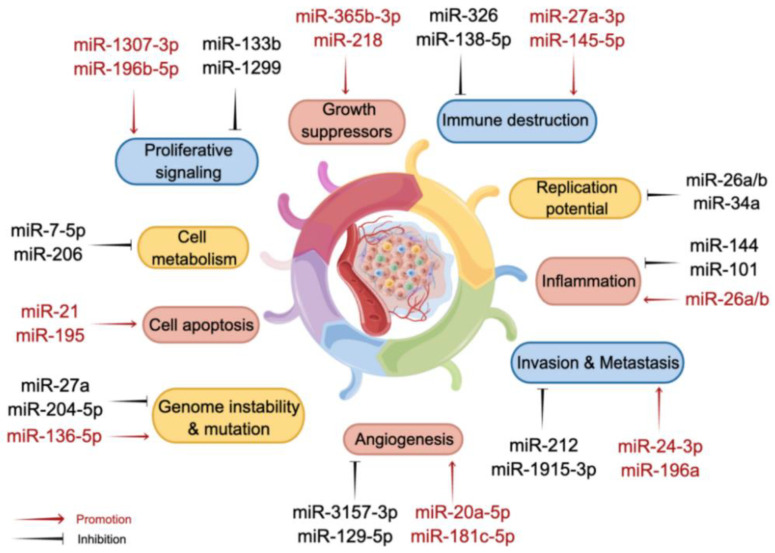
MiRNAs are involved in regulating cancer development through multiple mechanisms.

**Figure 3 biomolecules-13-00877-f003:**
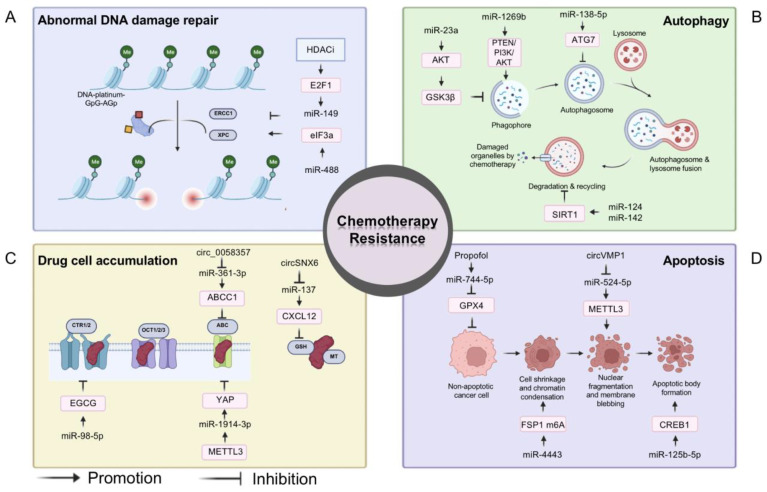
MiRNAs and chemotherapy. The mechanisms of chemotherapy resistance in tumors: (**A**) Abnormal DNA damage repair; (**B**) Removal of damaged organelles by autophagy; (**C**) Abnormal expression of drug transport proteins and consequently reduction of drug concentration; (**D**) Blocking apoptosis and ferroptosis. MiRNAs can cause or reverse chemotherapy resistance through the mechanisms described above.

**Figure 4 biomolecules-13-00877-f004:**
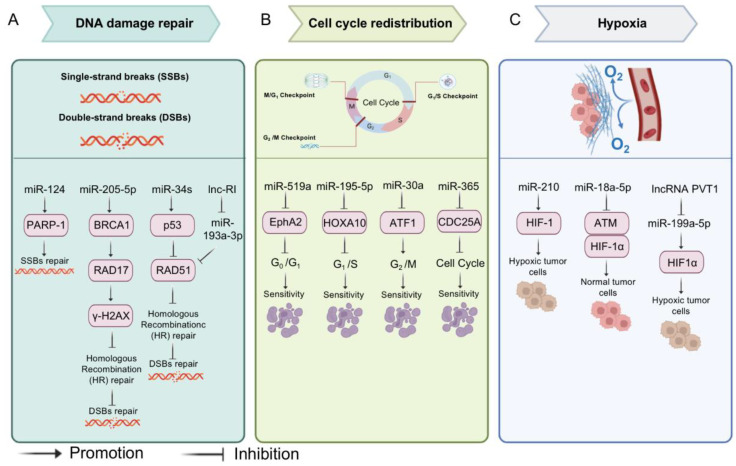
MiRNAs and radiotherapy. The mechanisms of radiotherapy resistance in tumors: (**A**) DNA double-strand breaks (DSBs); (**B**) Activation of cell cycle checkpoints; (**C**) Hypoxia. MiRNAs can cause or reverse radiotherapy resistance through the mechanisms described above.

**Figure 5 biomolecules-13-00877-f005:**
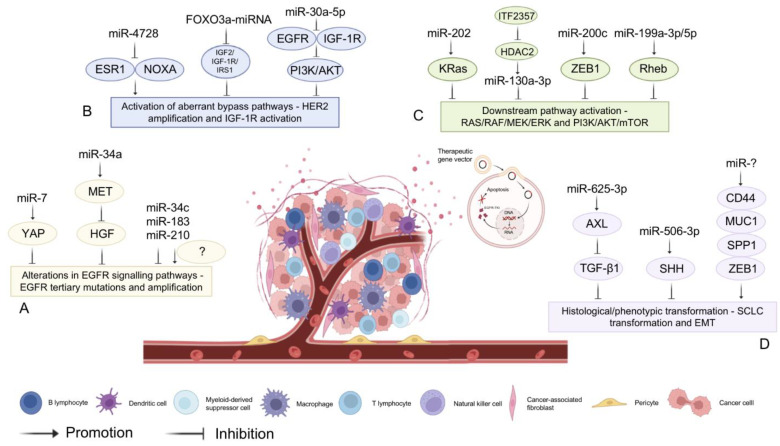
MiRNA-assisted targeted therapy. The mechanisms of targeted therapy resistance in tumors: (A) Alterations in EGFR signaling pathways; (B) Activation of aberrant bypass pathways—HER2 amplification and aberrant IGF-1R activation; (C) Downstream pathway activation—RAS/RAF/MEK/ERK and PI3K/AKT/mTOR; (D) Histological/phenotypic transformation—SCLC and EMT. MiRNAs can cause or reverse targeted therapy resistance through the mechanisms described above.

**Figure 6 biomolecules-13-00877-f006:**
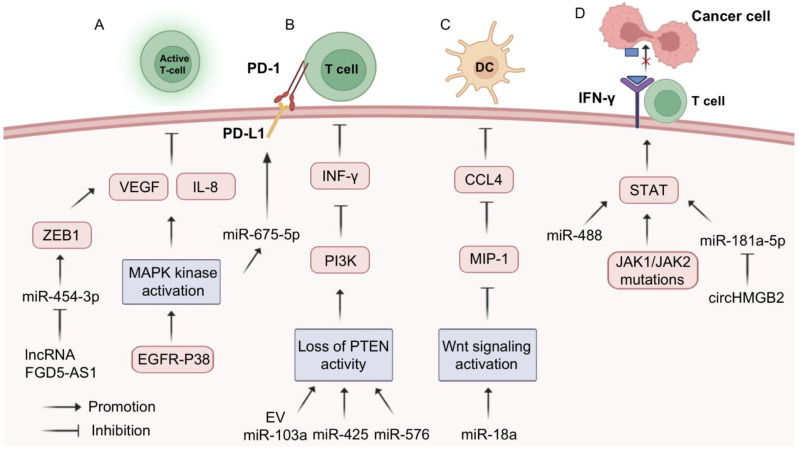
MiRNA-assisted immunotherapy. Alterations in intracellular-related signaling have four main effects on immunotherapy resistance: (A) Excessive activation of intracellular MAPK kinase pathway leads to IL-8 and VEGF production and inhibits T-lymphocyte infiltration into tumor tissue; (B) Loss of activity of the PTEN protein pathway leads to increased PI3K activity and reduced cytotoxic T-lymphocyte activity; (C) Over-activation of the Wnt signaling pathway leads to impaired or even suppressed production of MIP-1, which reduces infiltration of CD103+ dendritic cells (DCs) into tumor tissue; (D) Mutations in the JAK1/JAK2 gene promote the excessive activity of STATs family proteins and consequently the appearance of abnormal forms of IFN-γ receptors on the surface of tumor cells. MiRNAs can cause or reverse targeted therapy resistance through the mechanisms described above.

**Figure 7 biomolecules-13-00877-f007:**
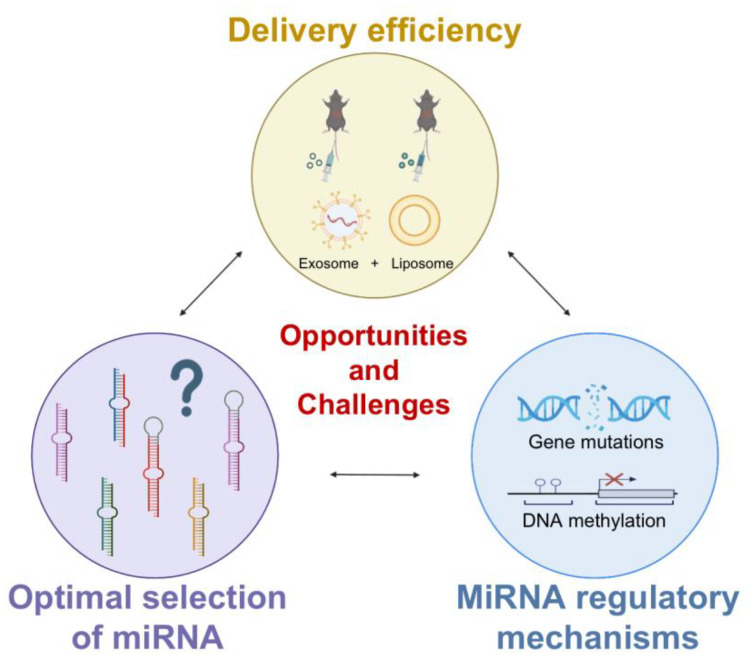
Opportunities and challenges in the use of miRNAs in clinical practice.

**Table 1 biomolecules-13-00877-t001:** MiRNA-based lung cancer therapies.

MiRNAs or lncRNAs	Expression	Cell Source	Target	Function	Ref.
MiRNA replacement therapies					
Let-7a	Down	A549	NIRF	Inhibit the proliferation	[24]
Let-7a	Down	A549	k-Ras c-Myc	Inhibit the growth	[25]
Let-7b-3p	Down	A549 PC9 H1299 etc.	BRF2	Inhibit the proliferation and metastasis	[26]
Let-7c	Down	SKMES-1 H520 H157 etc.	ITGB3 MAP4K3	Inhibit migration and invasion	[27]
Suppressive miRNA therapies					
LINC00336	Up	A549 H358	CBS	As a ceRNA of miR-6852 to inhibit the growth	[34]
MiR-21-5p	Up	A549	PDCD4	Promote apoptosis	[37]

**Table 2 biomolecules-13-00877-t002:** MiRNA-based preclinical studies and clinical trials.

MiRNAs	Target	Delivery Vehicle	Approach	Tumor Type	Results	Ref.
Preclinical Trials						
Let-7b	Kras	Adenovirus	Intranasal	NSCLC	66% reduction in orthotopic tumor burden; Reduced xenograft growth	[10]
MiR-29b	CDK6	Cationic lipoplexes	Caudal	NSCLC	60% xenograft growth inhibition	[152]
MiR-200c	PRDX2 GABP/Nrf2 SESN1	Amphoteric liposome	Subcutaneous	NSCLC	MiR-200c plus radiotherapy delayed xenograft growth	[153]
MiR-34a and Let-7b	Kras p53	Neutral lipid emulsion	Caudal	NSCLC	40% increased survival with combination or miR-34a alone	[28]
MiR-126	PTEN/PI3K/AKT	231-Exosome	Intravenous	NSCLC	Inhibit the formulation of lung metastasis	[154]
MiR-101	BCL6	AD-MSC-EVs	Caudal	Osteosarcoma	Inhibit the formulation of lung metastasis	[155]
MiR-21-5p	BTG2	UTMD	Caudal	NSCLC	Reduced the size and volume of xenograft growth	[156]
Clinical Trials						
MiR-16	EGFR	EDVs	Intravenous	NSCLC MPM	5 × 10^9^ TargomiRs once weekly was the maximum tolerated dose.	[160]
MiR-34	Kras P53 PDGFR CDK4 etc.	Liposome	Intravenous	Solid tumors Hematologic malignancies	Trial was closed early due to serious immune-mediated AEs	[12]

## Data Availability

Not applicable.

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
