# Peer review of "MiRNA-Based Therapies for Lung Cancer: Opportunities and Challenges?"

_biomolecules, 2023, doi:10.3390/biom13060877_

Round 1
Reviewer 1 Report
The authors give a good review of miRNAs in lung cancer and their potential utilities. Just a few comments:
- abbreviations are missing.
- The bibliography should be placed before the semicolons.
- In Figure 1, the numerical references are missing.
- In the figures, although it is understood to put what the arrows mean.
- Table summarizing point 2.
- Line 315, says numerous studies and does not refer to any. The same happens in line 384.
- In table 1 put which is preclinical and which is the clinical trial.
Reviewer 2 Report
The article by Yang at al. provides an overview of the recent data describing the role of microRNAs (miRNAs) as strategies for lung cancer therapy. They discuss the promises of miRNA-based therapies and miRNAs as a complementary strategy to the traditional cancer treatments. They further discuss the advances of using miRNA-based therapy in clinical practice.
The topic is important and the review has great potential to provide a valuable summary of recent findings about the role of miRNAs in lung cancer treatment. However, the manuscript needs a thorough revision before being published. While the review contains a plethora of knowledge, it is very difficult to follow up in reading.
The major concern is that the information in the text does not correspond with most of the figures (Figures 3, 4, 5, and 6). For example, miR-138-5p, miR-1914-3p, and miR-4443 from Figure 3 are not mentioned in the text. Also, most miRNAs from Figure 4 are not described in the text and opposite, miR-18a-5p, miR-126, miR-296, and miR-20b-5p discussed in the paragraph describing Figure 4 are not illustrated at the Figure 4.
Similarly, multiple miRNAs present in Figures 5 and 6, are not mentioned in the text and miRNAs in the text describing these Figures are not illustrated in these Figures.
The authors must revise and organize the correspondence of Figures and the text. The other option is to make the tables that would better guide the reader with references for each miRNA.
From all the miRNAs given in Table 1 the authors discuss only miR-16 and miR-34. Can the authors elaborate the discussion for some other miRNAs from this table?
Round 2
Reviewer 2 Report
The authors have sufficiently addressed the prior concerns and greatly improved the manuscript. Thus, the paper is suitable for publication.